# 'Patera syndrome' during the COVID-19 pandemic in the Canary Islands (Spain)

**Elena Pisos-Álamo**[1][☉], **Michele Hernández-Cabrera**[1], **Laura López-Delgado**[1], **Nieves Jaén-Sánchez**[1], **Christian Betancort-Plata**[1], **Carmen Lavilla Salgado**[1], **Laura Suárez-Hormiga**[1], **Marta Briega-Molina**[2], **Cristina Carranza-Rodríguez**[3][*], **Margarita Bolaños-Rivero**[4], **Araceli Hernández-Betancor**[4], **José-Luis Pérez-Arellano**[3][☉]

**1** Unit of Infectious Diseases and Tropical Medicine, University Insular Hospital of Gran Canaria and Department of Medical & Surgical Sciences, University of Las Palmas de Gran Canaria, Las Palmas de Gran Canaria, Spain, **2** Unit of Infectious Diseases and Tropical Medicine, University Insular Hospital of Gran Canaria, Las Palmas de Gran Canaria, Spain, **3** Research Institute of Biomedical and Health Sciences (IUIBS), University of Las Palmas de Gran Canaria, Las Palmas de Gran Canaria, Spain, **4** Microbiology and Parasitology Service, University Insular Hospital of Gran Canaria, Las Palmas de Gran Canaria, Spain

☉ These authors contributed equally to this work.
* cristina.carranza@ulpgc.es

**Data Availability Statement:** The Repository: this has been uploaded to the stable, open repository Zenodo (DOI: 10.5281/zenodo.13807911).

## Abstract

### Background

The number of migrants arriving on the shores of the Canary Islands continues to increase. The conditions under which the crossing is made, in small crowded, unsanitary boats (*pateras* or *cayucos*), have many and significant health problems.

### Objective

To describe the demographic, clinical, microbiological characteristics and evolution of a series of patients who recently arrived by *patera* and required hospitalization.

### Patients and methods

This observational, cross-sectional, and retrospective study included all patients newly arrived in Gran Canaria (Spain) by patera or cayuco from 2020 to 2022. Acute patera syndrome (APS) was defined as one or more of the following: dehydration, hypothermia, shock or rhabdomyolysis. Skin and soft tissue or musculoskeletal patera syndrome (SSTMSPS) was defined as conditions characterized by lesions of the skin, subcutaneous tissue, bone, or joint, excluding superficial erosions.

### Results

During the study period, 193 migrants were admitted, mostly males with a median age of 23 years from West Africa. A total of 36.99% presented with APS with a single diagnostic criterion (most commonly dehydration, 86.9%), 11.56% with SSTMPS and 51.44% with both syndromes. A total of 109 patients presented with SSTMSPS, the most common being lower extremity ulcers. The most frequently isolated microorganisms were gram-negative (i.e. *Shewanella algae*). The McMahon score effectively predicted the need for renal

**Funding:** The author(s) received no specific funding for this work.

**Competing interests:** The authors have declared that no competing interests exist.

replacement therapy in cases of rhabdomyolysis. Twenty patients presented with pneumo-mediastinum, which was benign. SARS-CoV-2 infection was not a problem in any of them. Surgical intervention was required in 22% of cases, including 8 amputations, all of which were minor. No patient died during admission.

## Conclusion

Patera syndrome has specific characteristics that should be identified promptly to initiate the most effective treatment for optimal outcomes.

## Introduction

For geographical reasons, Spain is a frequent destination for irregular migrants from Africa. The two main routes of entry are the Atlantic (to the Canary Islands) and the Mediterranean (by sea to Andalusia, and by land to the tiny Spanish enclaves of Ceuta and Melilla on the North African coast) (**Fig 1**). Regarding the Canary Islands, the province of Las Palmas has received the largest number of migrants.

Irregular migration by sea to the Canary Islands since the end of the 20th century has followed a biphasic pattern, with a first wave between 1997 and 2010, a sharp decrease between 2010 and 2017, and a progressive increase until now (**Fig 2**) [1]. In fact, the highest historical figure (39,910) was reached in the Canary Islands in 2023 [1]. This recent increase is partly related to border restrictions in the Maghreb due to the SARS-Cov-2 pandemic.

This journey is usually made in small boats called *pateras* (with a capacity of 10 to 20 people) or *cayucos* (with a capacity of 40 to 70 individuals) [2, 3]. The long duration of the voyage (which can last up to two weeks) and the large number of passengers, often exceeding the maximum capacity, create a series of unsuitable conditions that are responsible for the occurrence of health problems [2–5]. For example, the mobility of the passengers is restricted, both by the lack of space and sometimes by the use of ropes to prevent them from falling into the sea or the boat from capsizing. Passengers are also exposed to low temperatures, scarce supplies of fresh water and food, and inadequate hygiene conditions. As a result, migrants often suffer from malnutrition, ingest salt water or water contaminated with urine or feces, and/or develop skin and soft tissue and/or musculoskeletal injuries that are related to contact with biological fluids, rotting food, or fuel-water emulsions.

A significant number of migrants, although not precisely quantified, perhaps as many as one third, may die during the voyage [2, 3]. There is little information on the health problems of migrants who have recently arrived in Europe and the results vary according to the migration route (Canary Islands, Western or Central Mediterranean) and the date of the study [2, 6–8]. Fortunately, most of the irregular migrants arriving by sea do not require health care or present with minor health problems that can be treated on an outpatient basis. However, a small but numerically significant percentage require hospital admission [2, 6].

In our geographical area, two types of serious manifestations have been previously described in these patients: acute alterations such as hypothermia, dehydration, shock or rhabdomyolysis [9], and cutaneous-musculoskeletal lesions such as ulcers, tenosynovitis, fasciitis or osteomyelitis [4, 5].

Therefore, the aim of this study was to describe the demographic, clinical, microbiological characteristics and evolution of a large series of newly arrived 'patera' patients requiring hospital admission.

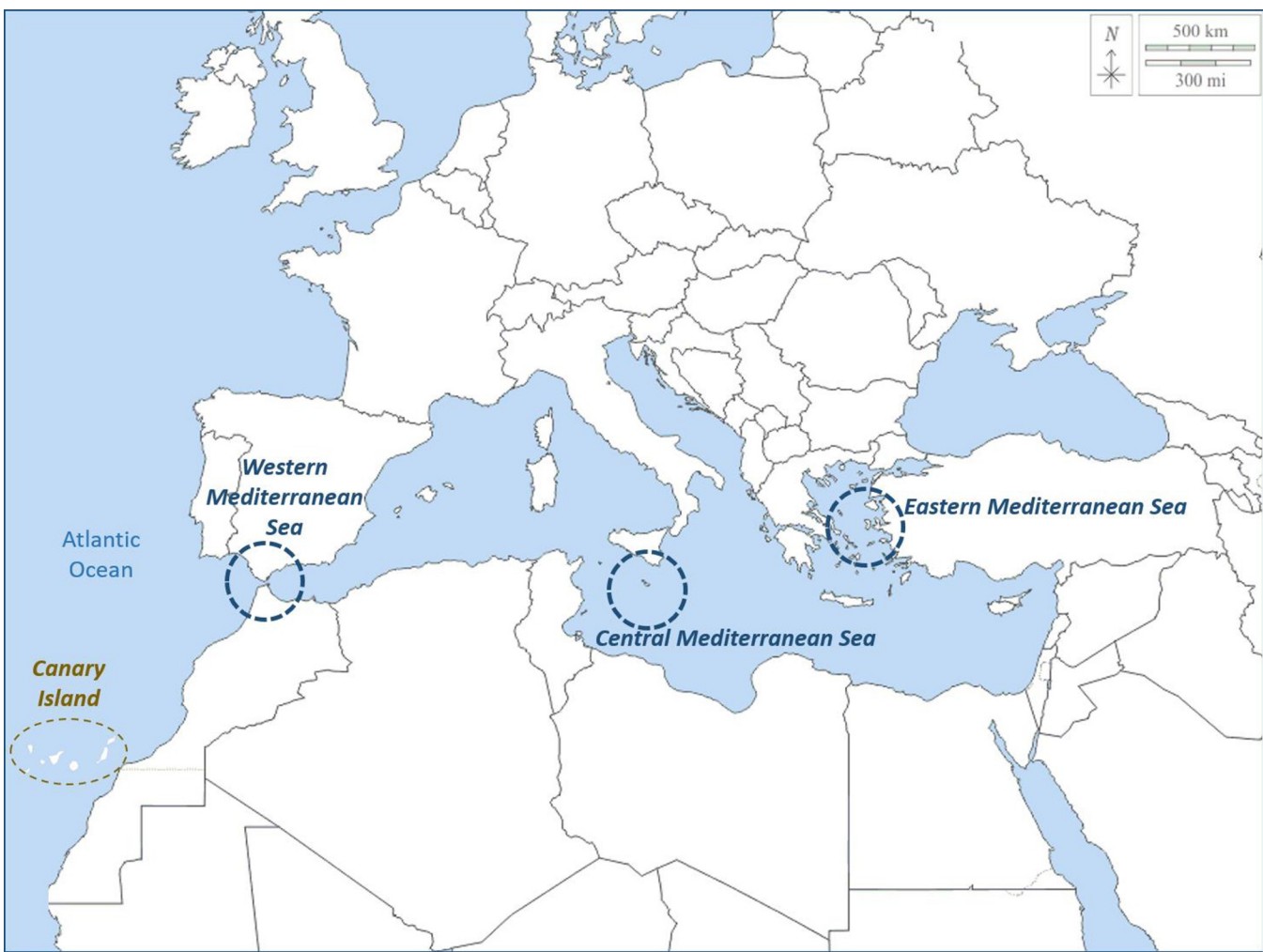

**Fig 1. Irregular migration routes to Europe.** Based on https://d-maps.com/carte.php?num_car=3138&lang=es.

## Patients and methods

### Study design, setting and subjects

This observational, cross-sectional, retrospective study was conducted at the Hospital Universitario Insular de Gran Canaria (Spain), which treats patients over 14 years of age. The study population included all hospitalized patients who had recently arrived in Gran Canaria (Spain) by small boat (patera or cayuco) between 1 January 2020 and 31 August 2022.

### Variables

The following data were collected for each patient: i) epidemiologic data, including age, sex, country and region of origin, year and month of arrival; ii) general clinical data at the time of admission, including temperature, blood pressure, heart rate and respiratory rate, and clinical signs of dehydration; iii) skin and soft tissue and/or musculoskeletal lesions at the time of admission; iv) other clinical manifestations not included above; v) laboratory tests (complete blood count (CBC), serum biochemical parameters and urinalysis) obtained during the patient's first visit to the emergency department and 72 h later; vi) chest X-ray, including

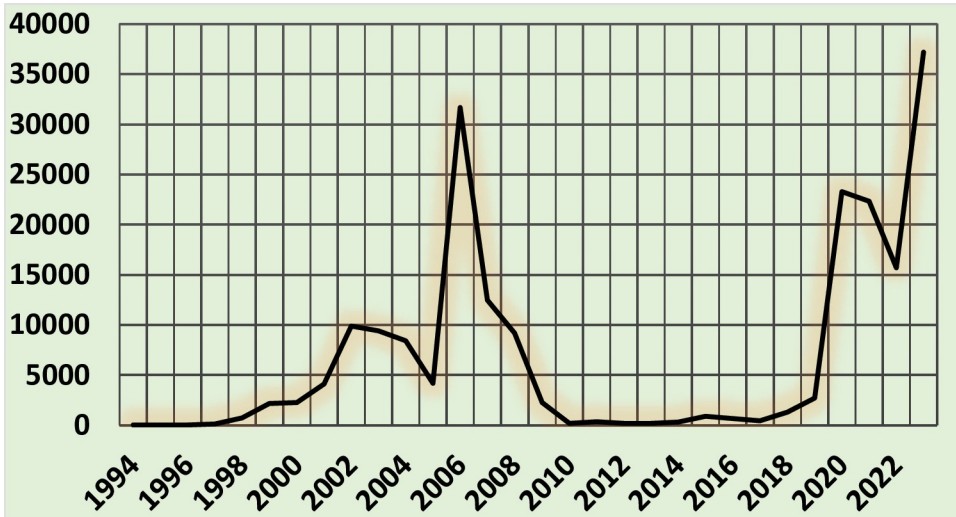

**Fig 2. Number of irregular migrants arriving in the Canary Islands by sea.**

specifically the presence of pneumomediastinum, subcutaneous emphysema, pneumothorax and pneumoperitoneum; vii) PCR detection of SARS-CoV2 using the Allplex 2019-nCoV assay (Seegene®, Seoul, South Korea), coproparasitic examination (Ritchie's Method and Kato-Katz tests), serological tests for HIV, HBV, HCV, and the treponemal test for syphilis using the chemiluminescent microparticle immunoassay (CMIA) (Abbot®). If the treponemal test was positive, the study was extended using the rapid plasma reagin (RPR) agglutination technique (Chronolab Systems®). The diagnosis of malaria was based on blood smear examination, immunochromatography (BinaxNOW Malaria, Abbott®) and multiplex real-time PCR (Altona Diagnostics®); viii) culture of skin and/or musculoskeletal lesions whenever possible; ix) blood, stool, and urine cultures in certain circumstances, and x) patient evolution, including the need for and type of surgical intervention for skin and/or musculoskeletal lesions, admission to intensive care (ICU), length of hospital stay and in-hospital mortality.

## Study definitions

The following definitions were used:

- *Acute patera syndrome* (APS): one or more of the following symptoms was noted: dehydration, hypothermia, shock or rhabdomyolysis. The presence of dehydration was based on classic physical signs and/or determination of a BUN/creatinine ratio greater than 25 [10, 11]. Patients with dehydration were classified into three groups: isotonic, hypertonic, or hypotonic, based on plasma osmolality (measured or calculated) [10, 11]. Hypothermia was considered when the body temperature was below 35˚ [12]. Shock was defined as the presence of at least three of the following: tachycardia ($> 100$ bpm), tachypnea ($\geq 22$/min), hypotension (mean arterial pressure less than 65 mm Hg) or metabolic acidosis (low HCO3 and low pH in plasma). Rhabdomyolysis was defined according to published criteria when CK (creatine kinase) activity exceeded the cut-off value of $> 1,000$ IU/L or CK $> 5 \times$ ULN [13, 14]. The severity of rhabdomyolysis was classified according to Farkas' criteria [15, 16] and the risk of acute renal failure according to the McMahon score [17].

- *Skin and soft tissue or musculoskeletal patera syndrome* (SSTMSPS): All patients with skin, subcutaneous tissue, bone or joint lesions were included and patients with superficial

erosions were excluded. Lesions (single or multiple) located in one or more anatomical regions were distinguished according to topography. The regions evaluated were the feet, other areas of the lower extremities, buttocks, sacrum, upper extremities and other areas (e.g. head, thorax, back or genitalia). The lesions were classified into five groups according to depth: cellulitis, ulcer (necrotic or purulent), abscess, fasciitis and osteoarticular involvement (tenosynovitis or osteomyelitis). Whenever possible, samples were obtained for microbiological study. In cases with positive culture, monomicrobial and polymicrobial forms were distinguished and the species involved were identified.

- *COVID-19*: WHO criteria were used to classify the severity of infection as mild (uncomplicated illness, including non-specific symptoms); pneumonia (no signs of severe pneumonia and no need for supplemental oxygen); severe pneumonia (fever or suspected respiratory infection plus one of the following: respiratory rate $> 30$ breaths/min; severe respiratory distress or SpO2 93% on room air) [18].

## Statistical analysis

Data analysis was performed using the DATAtab Online Statistical Calculator for Mac®. There were some missing data in the patient notes, which was noted in the analyses. Normality of data was assessed by the Kolmogorov-Smirnov test and homogeneity of variance by Levene's test.

Categorical data were presented as frequencies and percentages and continuous data as means and standard deviations (SD) or median and interquartile range (IQR), as appropriate.

Categorical variables were compared using Pearson's chi-squared test or Fisher's exact test, when indicated. Student's t test or ANOVA were used to compare two or more normally distributed continuous independent variables, while the Mann-Whitney or Kruskal Wallis tests were used as nonparametric tests. When significant differences were detected by ANOVA or Kruskal-Wallis tests, post-hoc tests (Bonferroni or Dunn-Bonferroni) were used. The Pearson or Spearman correlations were used to evaluate correlations between variables, depending on the sample distribution.

Statistical significance was determined using a two-sided p-value of less than 0.05.

## Ethical aspects

The study was conducted in accordance with the protocol and principles of the current revised version of the Declaration of Helsinki (Fortaleza, October 2013) and approved by the Ethics Committee, protocol number 2024-070-1.

The data and clinical images were collected as part of routine care by the responsible clinical team and were anonymized at the point of extraction. None of the clinical images allowed patient identification. Patient consent was not obtained due to the retrospective nature of the study.

## Results

### Demographic data

A total of 193 patients were included: 35 were women (5 of whom were pregnant). **Table 1** shows the age of the migrants according to sex. The statistical study showed significant differences between males and females, with age being higher in females. Most of the patients came from West Africa, followed by North Africa (**Table 1**). In terms of country of origin, most patients came from Mali, followed by Morocco and Côte d'Ivoire).

**Table 1. Epidemiologic data.**

| | | Male | Female | Total |
|---|---|---|---|---|
| **Number** (percentage) | | 158 (89) | 35 (11) | 193 (100%) |
| **Age**\*, Mean (SD) | | 23.69 (5.96) | 28.26 (5.04) | 24.52 (6.05) |
| **Geographical area**\*\* | **Country**\*\* | | | |
| North Africa | | 41 | 0 | 41 |
| | Morocco | 41 | 0 | 41 |
| West Africa | | 101 | 29 | 130 |
| | Burkina Faso | 2 | 0 | 2 |
| | Cameroon | 1 | 0 | 1 |
| | Central Africa | 1 | 0 | 1 |
| | Côte d'Ivoire | 6 | 25 | 31 |
| | Gambia | 9 | 1 | 10 |
| | Ghana | 1 | 0 | 1 |
| | Guinea | 5 | 3 | 8 |
| | Guinea Bissau | 1 | 0 | 1 |
| | Liberia | 1 | 0 | 1 |
| | Mali | 54 | 0 | 54 |
| | Mauritania | 2 | 0 | 2 |
| | Nigeria | 1 | 0 | 1 |
| | Senegal | 19 | 0 | 19 |

\* $p < 0.001$

\*\* Data were not available for 21 patients

Fifty-eight patients arrived in 2020, 125 in 2021 and 10 in 2022. The distribution of admissions according to month is shown in **Fig 3**, with the majority being between August and December.

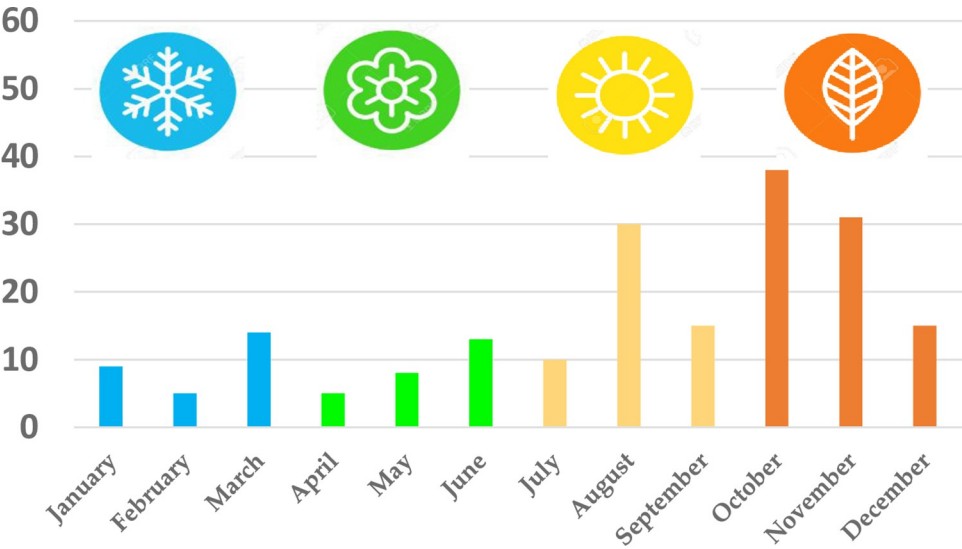

**Fig 3. Distribution of arrivals by month.**

## Clinical syndromes

Some 36.99% of patients presented with APS with a single diagnostic criterion, 11.56% with SSTMSPS without APS and 51.44% with both syndromes (Table 2); 20 patients had criteria for none of these syndromes. In the latter cases, the final diagnosis was bronchopneumonia (1), lobar pneumonia (2), obstetric complications (3), *Plasmodium falciparum* infection (1), pneumo-mediastinum (2), pneumothorax and tuberculosis (1). A final diagnosis could not be made in 9 patients.

Among the diagnostic criteria of APS, the most common was the presence of dehydration (86.9%), followed by rhabdomyolysis (71.9%), hypothermia (16.3%) and shock (5.8%). According to the plasma osmolality data, 27% had a hypertonic form ($> 295$ mOsm/kg), 31.5% an isotonic form (285–295 mOsm/kg) and another 31.5% a hypotonic form ($< 285$ mOsm/kg). The presence of rhabdomyolysis was documented in 110 patients: 69 were mild (62.7%), 33 moderate (30%) and 8 severe (7.3%). Although less common, hypothermia was significantly associated with cases of APS with concomitant skin, soft tissue, and musculoskeletal involvement.

Superficial skin involvement is practically universal in migrants arriving by boat. Less frequently, deeper layers of the skin, soft tissues or musculoskeletal structures are involved, which was observed in 109/193 (56%) of patients in this series (Fig 4).

Table 3 shows the anatomical regions involved and the type of lesions according to their depth. In 63 patients, a single region was involved, and in 46 patients, multiple (two or more) areas. In terms of depth, most lesions were ulcers (purulent, necrotic, or mixed), with cellulitis being another common manifestation.

With the exception of cellulitis, the rest of the lesions were cultured. Of 109 cultures performed, 49 were negative, 18 were monomicrobial and 38 were polymicrobial. In the

**Table 2. Clinical differences between the main types of 'Patera syndrome'.**

| | APS* | SSTMSPS* | APS + SSTMSPS* | P value |
|---|---|---|---|---|
| **Number** (percentage) | 64/173 (36.99) | 20/173 (11.56) | 89/173 (51.44) | |
| **Male/Female** ratio** | 55 /9 (6:1) | 15/5 (3:1) | 71/18 (4:1) | 0.45 |
| **Age (years),** Mean (SD)*** | 25 (7) | 26 (8) | 24 (5) | 0.31 |
| **Geographic area** ** | | | | 0.11 |
| North Africa | 19/56 | 5/19 | 14/77 | |
| Western Africa | 37/56 | 14/19 | 63/77 | |
| **Criteria for APS** ** | | | | |
| Dehydration | 54/64 | - | 79/89 | 0.46 |
| Hypothermia | 6/64 | - | 19/89 | **0.04** |
| Shock | 3/64 | - | 6/89 | 0.63 |
| Rhabdomyolysis | 43/64 | - | 67/89 | 0.27 |
| **SSTMSPS** ** | | | | 0.78 |
| One anatomical region | - | 11/20 | 52/89 | |
| >1 anatomical region | - | 9/20 | 37/89 | |
| **Average hospital stay (days)*** Median (IQ range) | 7 (4–12) | 9 (6–15) | 11(6–25) | **0.02** |
| **Intensive Care Unit (ICU) admission** ** | 1/64 | 7/20 | 4/89 | **< 0.01** |

* APS: Acute Patera Syndrome; SSTMSPS: Skin and Soft Tissue or Musculoskeletal Patera Syndrome

**$\chi^2$

***ANOVA test

**** Kruskal-Wallis test.

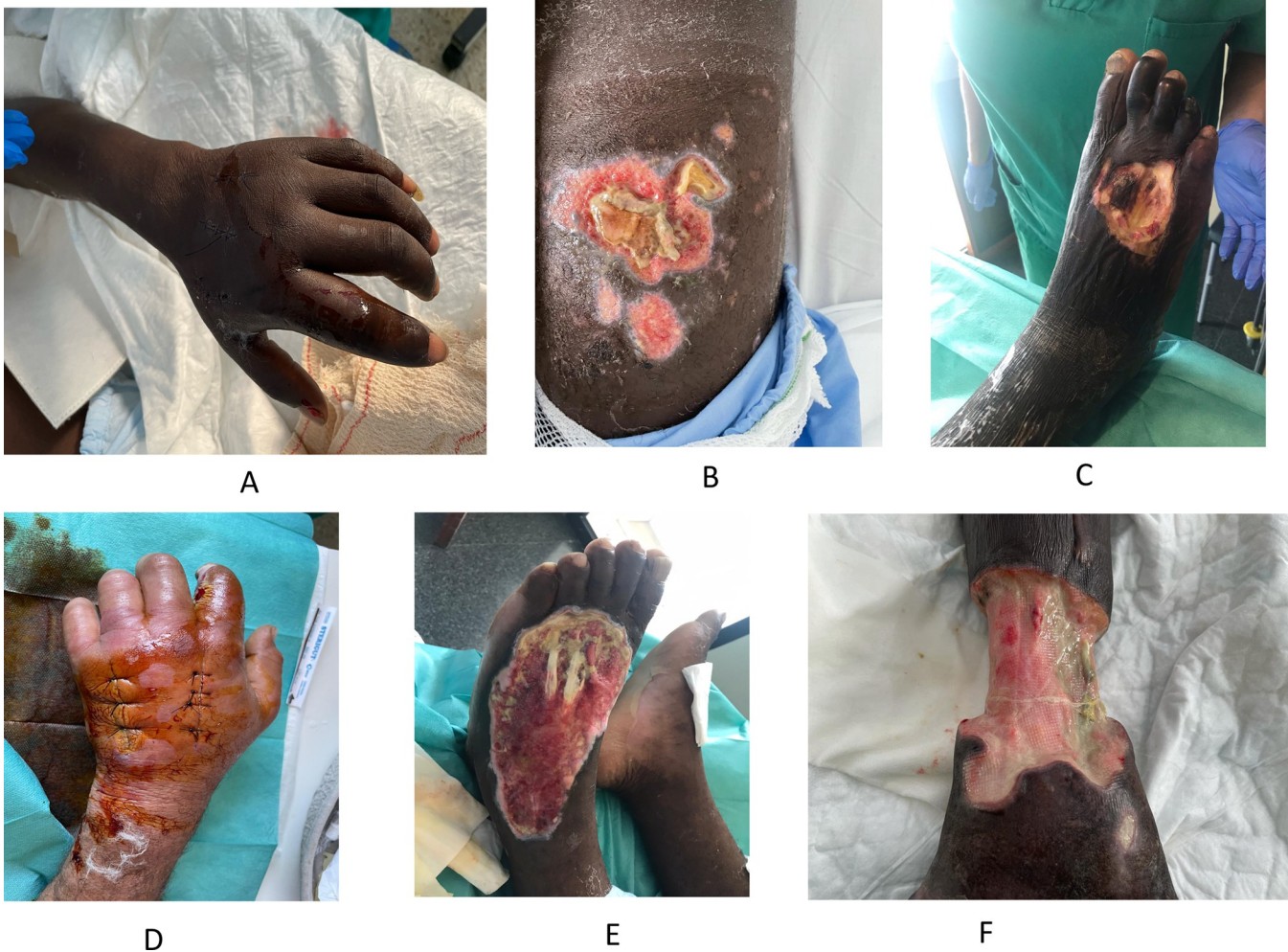

**Fig 4. Morphologic types of SSTMSPS.** A: Cellulitis; B: Purulent ulcers; C: Necrotic ulcer; D: Fasciitis; E: Tenosynovitis; F: Osteomyelitis.

remainder, and excluding contaminants, 99 different microorganisms were identified, as shown in **Table 4**.

## Analytical data

Several sources of analytical data were used in the initial definition of APS (time 0), as mentioned above. However, after performing the initial therapeutic maneuvers (mainly rehydration), noteworthy differences were observed (see Repository). **Table 5** shows only the main data from the laboratory studies obtained 72 hours after admission to the emergency department, following stabilization of the patient.

## Radiology

In addition to other specific data (i.e. lobar pneumonia, bronchopneumonia, tuberculosis or hydatidosis), 20 patients (10.3% of the total) presented radiological data suggestive of extrapulmonary air not related to trauma or iatrogenesis (**Fig 5**). The most common manifestation was pneumomediastinum (20 patients) associated with subcutaneous emphysema in 8 cases, and

**Table 3. Clinical differences between the main types of patera syndrome.**

| | Number | Percentage |
|---|---|---|
| **Anatomical region** | | |
| Feet | 30 | 18.40 |
| Lower limbs (not feet) | 46 | 28.22 |
| Buttocks | 33 | 20.25 |
| Sacrum | 6 | 3.68 |
| Upper limbs | 38 | 23.31 |
| Other | 10 | 6.13 |
| Total | **163** | **100** |
| **Depth of injury** | | |
| Cellulitis | 23 | 14.11 |
| Ulcer/s | 121 | 74.23 |
| Abscess | 5 | 3.07 |
| Fasciitis | 6 | 3.68 |
| Musculoskeletal | 8 | 4.91 |
| **Total** | **163** | **100** |

pneumothorax or pneumoperitoneum in two different patients. There was no statistically significant association between the presence of extrapulmonary air and SARS-CoV2 infection.

## Microbiological studies

Thirty-seven of the 193 patients (19%) were infected with SARS-CoV2: 21 were asymptomatic and 16 had mild COVID-19. Of 178 patients evaluated, 4 were infected with HIV. Eight of 154 migrants had a positive syphilis (treponemal) test, all of which were RPR negative. Regarding primary hepatotropic virus serology, 32 of 141 had data of past HBV infection (HBcAb + HBAg -) and 20 of 141 had current infection (HBcAb + HBAg +). None of the patients were HCV-infected.

## Other diseases

In addition to the above data, the presence of other diseases was evaluated and classified as acquired before the trip; acquired before or during the trip; and complications during admission. Among the pre-trip diseases, infectious causes [n = 7; *Echinococcus granulosus* (1), *Mycobacterium tuberculosis* (2), *Plasmodium falciparum* (3) and *Strongyloides stercoralis* (1)] and non-infectious causes [n = 5; Behçet's disease (1), rheumatic heart disease, sickle cell disease (1) and type 1 diabetes mellitus (2)] were distinguished. The main diseases acquired before and/or during the trip were: *Giardia duodenalis* infection (11), *Salmonella enterica* subsp. *enterica* infection (7), scabies (5) and respiratory pathogens [n = 5; *Streptococcus pneumoniae* (2), rhinovirus/enterovirus (2) and influenza B virus (1). Complications during hospitalization were nosocomial pneumonia (4), catheter-related bloodstream infection (4), pulmonary embolism (2), urinary tract infection (2), perforation of the gastrointestinal tract (1), subarachnoid hemorrhage (1) and postinfectious glomerulonephritis (1).

## Evolution

The initial empiric treatment given is shown in **Table 6**. It was later modified in accordance with the microbiological data obtained. At least one surgical procedure was required in 24 of the 109 patients with SSTMSPS (22%) (**Table 7**). Debridement and cleaning were performed

**Table 4. SSTMSPS (Skin and soft tissue or musculoskeletal patera syndrome) microbiological data.**

| **Facultative anaerobic and obligate aerobic bacteria** | | | | | |
|---|---|---|---|---|---|
| **Gram stain** | **Order** | **Family** | **Genus** | **Species** | **Number** |
| **Positive** | | | | | **42** |
| | *Corynebacteriales* | *Corynebacteriaceae* | *Corynebacterium* | *C. diphtheriae* | 1 |
| | *Lactobacillales* | *Enterococcaceae* | *Enterococcus* | *E. faecalis.* *E. raffinosus* *E. thailandicus* | 3 |
| | *Bacillales* | *Staphylococcaceae* | *Staphylococcus* | *S. aureus* | 25 |
| | *Lactobacillales* | *Streptococcaceae* | *Streptococcus* | *S. agalactiae* | 3 |
| | | | | *S. dysgalactiae* | 4 |
| | | | | *S. pyogenes* | 6 |
| **Negative** | | | | | **48** |
| | *Aeromonadales* | *Aeromonadaceae* | *Aeromonas* | *A. hydrophila* | 1 |
| | *Enterobacterales* | *Enterobacteriaceae* | *Enterobacter* | *E. cloacae* | 1 |
| | | | *Escherichia* | *E. coli* | 5 |
| | | | *Klebsiella* | *K. aerogenes* | 2 |
| | | | | *K. pneumoniae* | |
| | | *Morganellaceae* | *Morganella* | *M. morganii* | 8 |
| | | | *Proteus* | *P. mirabilis* | 3 |
| | | | | *P. vulgaris* | |
| | | | *Providencia* | *P. stuarti* | 2 |
| | *Pseudomonadales* | *Pseudomonadaceae* | *Pseudomonas* | *P. aeruginosa* | 7 |
| | | | | *P. hauseri* | 1 |
| | *Alteromonadales* | *Shewanellaceae* | *Shewanella* | *S. algae* | 13 |
| | *Vibrionales* | *Vibrionaceae* | *Vibrio* | *V. alginolyticus* | 5 |
| **Obligate anaerobic bacteria** | | | | | |
| **Gram stain** | | | | | |
| **Positive** | | | | | **1** |
| | *Clostridia* | *Peptoniphilaceae* | *Finegoldia* | *F. magna* | 1 |
| **Negative** | | | | | **6** |
| | *Bacteroidales* | *Bacteroidaceae* | *Bacteroides* | *B. fragilis* | 5 |
| | *Fusobacteriales* | *Fusobacteriaceae* | *Fusobacterium* | *F. varium* | 1 |
| **Yeasts** | | | | | |
| | *Saccharomycetales* | *Saccharomycetaceae* | *Candida* | *C. tropicalis* | 2 |

in all cases, reconstructive plastic surgery in 11 patients and partial limb amputation in 8 patients. All were minor, according to Ratliff's classification [19] (6 trans phalangeal, 1 Chopart, 1 Pirogoff). ICU admission was required in 12 patients, with significant differences between clinical syndromes, being more frequent in patients with SSTMSPS (**Table 2**). The overall median stay was 9 days (IQR: 11) with a significant association between longer stay and the presence of both syndromes (APS and SSTMSPS) (**Table 2**). None of the patients died during hospital admission.

## Discussion

Since 1997, there has been a steady increase in the number of people arriving irregularly in Spain in small boats or cayucos. In general, the Canary Islands are not their intended destination but a country of transit on their way to the rest of Europe. For this reason, the correct

**Table 5. Analytical data at 72 hours.**

| | All Patients | Acute Patera Syndrome (APS) | Skin and Soft Tissue or Musculoskeletal Patera Syndrome (SSTMSPS) | APS plus SSTMSPS | P value |
|---|---|---|---|---|---|
| **Anemia** | 92/162 (56.8%) | 28/50 | 9/18 | 47/77 | 0.84 |
| **Mean corpuscular volume MCV (fl)** | | | | | 0.75 |
| Decreased (< 80) | 31/192 (16,7%) | 9/50 | 3/20 | 18/89 | |
| Normal (80–100) | 158/192 (82.3%) | 39/50 | 17/20 | 71/89 | |
| Elevated (> 100) | 3/192 (1.6%) | 2/50 | 0/20 | 0 | |
| **Total leukocytes /μL** | | | | | 0.90 |
| Decreased (< 4,000) | 111/158 (70.2%) | 32/50 | 11/19 | 52/77 | |
| Normal (4.000–11.000) | 41/158 (25.9%) | 14/50 | 7/19 | 20/77 | |
| Elevated (> 11.000) | 6/158 (3.8%) | 4/50 | 1/19 | 5/77 | |
| **Eosinophils/μL** | | | | | 0.99 |
| Elevated (> 450) | 19/163 (11.6%) | 5/50 | 2/18 | 9/77 | |
| **Platelets (x $10^3$/μL)** | | | | | 0.20 |
| Decreased (<150) | 19/193 (11.6%) | 4/50 | 1/18 | 14/77 | |
| Normal (150–400) | 158/193 (82.4%) | 38/50 | 17/18 | 58/77 | |
| Elevated (> 400) | 16/193 (8,3%) | 8/50 | 0/18 | 5/77 | |
| **Plasma creatinine level (mg/dL)** | | | | | 0.4 |
| Decreased (< 0.8) | 119/163 (73,0%) | 35/52 | 11/16 | 61/77 | |
| Normal (0.8–1.2) | 42/163 (25.7%) | 16/52 | 5/16 | 15/77 | |
| Elevated (> 1.2) | 2/163 (1.3%) | 1/52 | 0/16 | 1/77 | |
| **$Na^+$ (mEq/L)** | | | | | 0.23 |
| Decreased (< 135) | 23/155 (14.8%) | 12/47 | 1/16 | 10/75 | |
| Normal (135–145) | 123/155 (79.3%) | 33/47 | 15/16 | 58/75 | |
| Elevated (> 145) | 9/155 (5.8%) | 2/47 | 0/16 | 7/75 | |
| **$K^+$ (mEq/L)** | | | | | 0.74 |
| Decreased (< 3,5) | 5/155 (3.2%) | 1/47 | 0/16 | 3/75 | |
| Normal (3,5–5,0) | 147/155 (94.8%) | 45/47 | 15/16 | 71/75 | |
| Elevated (> 5,0) | 3/155 (2.0%) | 1/47 | 1/16 | 1/75 | |
| **AST (U/L)** | | | | | **<0.01** |
| Normal (0–35) | 35/192 (18.2%) | 5/63 | 11/20 | 8/89 | |
| Elevated (36–108) | 157/192 (81.8%) | 29/63 | 6/20 | 37/89 | |
| Very high (>108 UNL) | 75/193 (38.8%) | 29/63 | 3/20 | 44/89 | |
| **ALT (U/L)** | | | | | 0.13 |
| Normal (0–44) | 96/192 (50%) | 30/64 | 15/20 | 38/89 | |

*(Continued)*

**Table 5.** (Continued)

| | All Patients | Acute Patera Syndrome (APS) | Skin and Soft Tissue or Musculoskeletal Patera Syndrome (SSTMSPS) | APS plus SSTMSPS | P value |
|---|---|---|---|---|---|
| Elevated (45–135) | 96/192 (50%) | 29/64 | 4/20 | 43/89 | |
| Very high (>135 UNL) | 14/192 (7.3%) | 5/64 | 1/20 | 8/89 | |
| **LDH (U/L)** | | | | | 0.31 |
| Normal (0–247) | 35/192 (18.2%) | 11/61 | 5/20 | 8(89 | |
| Elevated (248–744) | 157/192 (81.8%) | 45/61 | 13/20 | 72/89 | |
| Very high (> 744 UNL) | 16/192 (8.3%) | 5/61 | 2/20 | 9/89 | |

*$\chi^2$ test

term for these people is *migrants*, as opposed to *immigrants*, which implies that they intend to stay in the destination country.

During the period studied, 56,224 migrants arrived by sea in the Canary Islands, of whom 193 (0.34%) required hospitalization. Although we do not have exact data on the number of patients who arrived specifically in Gran Canaria, the hospitalization rate would have been less than 0.34% (193/56,224). These figures are similar to those reported in other published studies, which indicate that the need for hospital care is generally low (2.4–2.9%) [2, 8]. It should be noted however that this figure is sometimes overestimated, as newly arrived migrants are referred without a clear reason due to the language barrier, which hinders their ability to express their symptoms [6].

Regarding age, both in our study and in the various published series [4, 6–8], it is more common to find young men under 25 years of age, mostly from sub-Saharan Africa. Given that they migrate in search of work and a better quality of life, the most common age group, not surprisingly, is between 18–40 years [8]. However, as our data and those of other studies show, the number of women and children has gradually increased over the years [20]. In this series, 35 women were included, 5 of whom were pregnant (2.59%), a lower figure than that reported by other authors (7.2%) [8]. On the other hand, the age of the women included in the study was significantly higher than that of the men.

The geographical origin of the patients in our series is variable, although the majority came from Mali, Morocco, and Côte d'Ivoire. These data are very different from those reported in other publications on migrants arriving by boat in the south of the Iberian Peninsula (from Algeria and Morocco) [8, 20] and in Italy (from Eritrea, Nigeria and Somalia) [7]. The migration route followed is the most likely explanation for these data (**Fig 1**). It should be noted that in our study practically all the women came from Côte d'Ivoire, which is explained by the violent conditions, sexual exploitation of women and genital mutilation [21].

Although all the reviewed publications indicate that the annual distribution of migrants is greater between the months of July and September [2, 6, 8], in our series, the highest number of cases occurred between the months of October and November. This may be attributed to climate change, which has shifted favorable weather conditions towards the end of the year [2]. In addition, the average water temperature in the part of the Atlantic where the Canary Islands are located is 21°C, which may favor infection with certain microorganisms, such as *Shewanella algae*, that are associated with coastal areas and warm temperatures [22] and contribute to the lower incidence of hypothermia (see below).

The reasons for medical care in irregular migrants vary widely between studies. For example, in some series, the most frequent cause was febrile syndrome (mainly malaria and

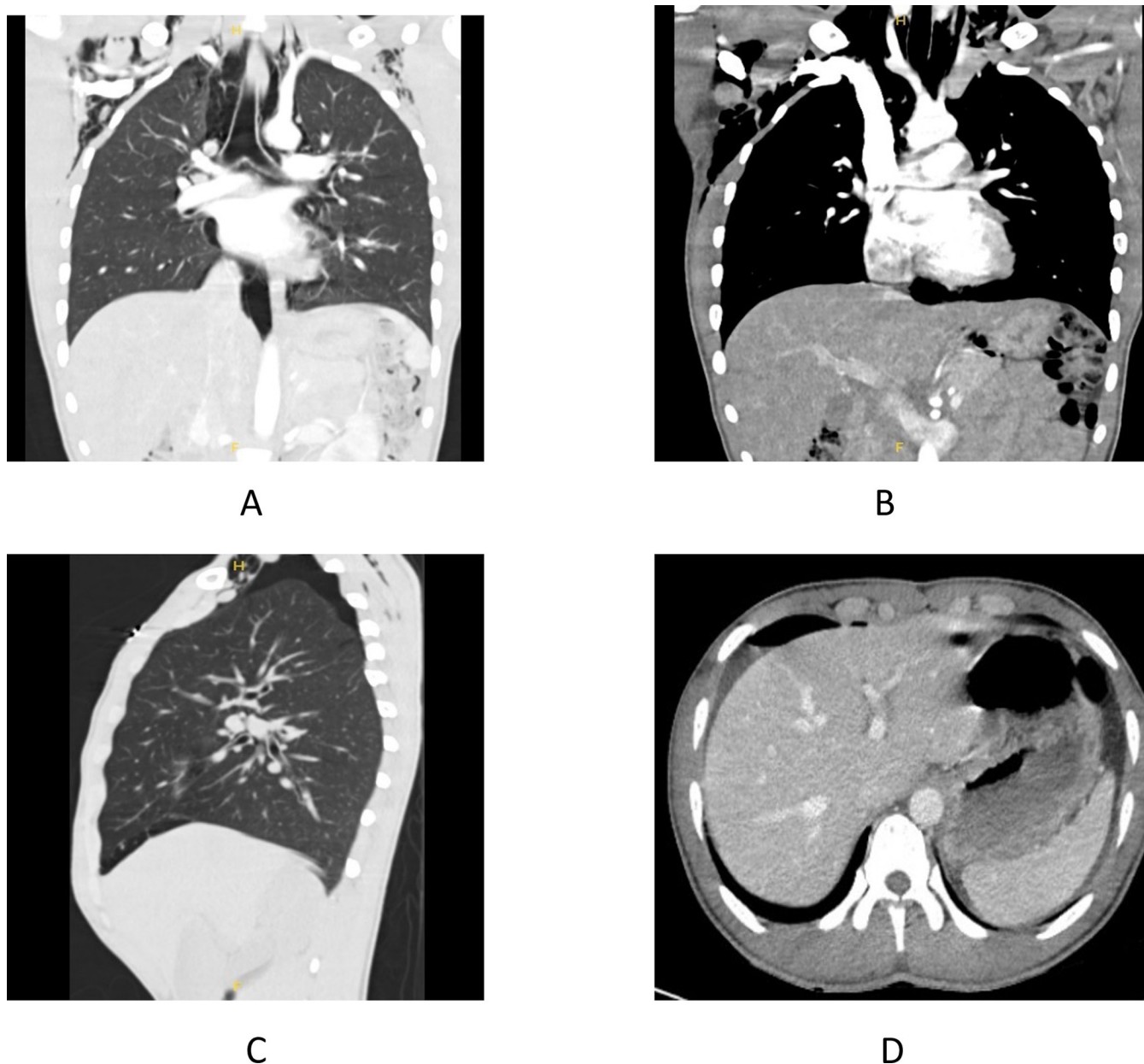

**Fig 5. Extrapulmonary air.** A: Pneumomediastinum; B: Subcutaneous emphysema; C: Pneumothorax; D. Pneumoperitoneum.

tuberculosis) [6]; in another study, it was skin lesions (scabies or chickenpox) [7] and in other publications, skin and soft tissue lesions associated or not with dehydration [2, 8]. In our study, most of the patients were included in the previously defined syndromes: APS 36.9%, SSTMSPS 11.56% and both 51.44%. The main manifestations of acute patera syndrome were dehydration and rhabdomyolysis; the presence of hypothermia or shock were less frequently observed. The definitive diagnosis of dehydration is a complex process in clinical practice and depends on physical examination data and/or use of analytical studies (mainly plasma osmolality or natremia) [10, 11, 23]. It is not uncommon therefore for data obtained from different studies to give discrepant results. This study found that approximately one third of dehydrated

Table 6. Empiric antimicrobial treatment of SSTMSPS (Skin and soft tissue or musculoskeletal patera syndrome).

| Empiric treatment | n | % |
|---|---|---|
| **Amoxicillin/clavulanate-based** | 79 | 74.53 |
| Amoxicillin/clavulanate *plus* Doxycycline | 46 | 43.40 |
| Amoxicillin/clavulanate | 29 | 27.36 |
| Amoxicillin/clavulanate *plus* Clindamycin | 4 | 3.77 |
| **Carbapenem-based** | 11 | 10.38 |
| Meropenem *plus* Doxycycline | 4 | 3.77 |
| Meropenem | 3 | 2.83 |
| Meropenem *plus* Doxycycline *plus* Linezolid | 1 | 0.94 |
| Imipenem | 2 | 1.89 |
| Ertapenem *plus* Ciprofloxacin | 1 | 0.94 |
| **Piperacillin/Tazobactam-based** | 8 | 7.55 |
| Piperacillin/Tazobactam | 3 | 2.83 |
| Piperacillin/Tazobactam *plus* Doxycycline | 2 | 1.89 |
| Piperacillin/Tazobactam *plus* Daptomycin | 1 | 0.94 |
| Piperacillin/Tazobactam *plus* Doxycycline *plus* Linezolid | 1 | 0.94 |
| Piperacillin/Tazobactam *plus* Linezolid | 1 | 0.94 |
| Piperacillin/Tazobactam *plus* Levofloxacin | 1 | 0.94 |
| **Others** | 7 | 6.60 |
| Ciprofloxacin *plus* Clindamycin | 2 | 1.89 |
| Levofloxacin | 1 | 0.94 |
| Levofloxacin *plus* Daptomycin | 1 | 0.94 |
| Ceftriaxone | 2 | 1.89 |
| Cefazolin | 1 | 0.94 |
| **TOTAL** | 106 | |

patients had one of the major types: hypertonic, isotonic, or hypotonic. Factors involved in dehydration include inadequate fluid intake, ingestion of salt water or excessive fluid intake on arrival. Although studies of the effects of dehydration have primarily focused on the elderly [10, 11], it has been observed that as dehydration progresses, the body loses its ability to regulate temperature. This may contribute to hypothermia in these patients, as well as the appearance of wounds and delayed healing, contributing to the development of skin and soft tissue lesions. According to the established biochemical criteria (CK activity), approximately three-quarters of the admitted patients had rhabdomyolysis. However, it is important to note that these data may be due to ethnic factors (higher proportion of type IIa fibers and a lower proportion of type I in black individuals) not related to muscle mass or physical activity [24]. In fact, although 40% had moderate or severe rhabdomyolysis, only a minority showed elevated plasma creatinine or serum potassium. In this study, the McMahon score was calculated on admission to predict the risk of renal failure requiring renal replacement therapy (RRT) or mortality in patients with rhabdomyolysis [14]. The score was less than 6 in all patients. The incidence of hypothermia was lower than would be expected for a sea crossing in these conditions, which may be related to the warm temperature of the waters round the Canary Islands and the duration of the crossing. On the other hand, hypothermia was more frequent in cases in which APS and SSTMSPS coexisted, possibly due to the severity of the patient [12]. Skin, soft tissue or musculoskeletal injuries of varying depth and severity are very common due to the conditions of the voyage and the restraints used on boats to prevent capsizing. In contrast to the reviewed series, in which the frequency of skin and soft tissue infections was less than

Table 7. Surgical management of SSTMSPS (Skin and soft tissue or musculoskeletal patera syndrome).

| N | Surgical debridement/drainage | Reconstructive plastic surgery | Amputation | Others | Location |
|---|---|---|---|---|---|
| 1 | Yes | No | No | No | Hand |
| 2 | Yes | No | No | No | Sacral |
| 3 | Yes | No | No | No | Leg |
| 4 | Yes | No | No | No | Hand |
| 5 | Yes | Skin graft | No | No | Hand |
| 6 | Yes | Skin graft | No | No | Foot |
| 7 | Yes | Skin graft | No | No | Ankle |
| 8 | Yes | Skin graft | No | No | Leg & Thigh |
| 9 | Yes | Skin graft | No | No | Foot |
| 10 | Yes | Skin flap | No | No | Foot |
| 11 | Yes | Skin flap | No | No | Foot |
| 12 | Yes | No | Trans phalangeal | No | Hand |
| 13 | Yes | No | Trans phalangeal | No | Hand |
| 14 | Yes | No | Trans phalangeal | No | Hand |
| 15 | Yes | No | Trans phalangeal | No | Hand |
| 16 | Yes | Skin flap | Trans phalangeal | No | Hand |
| 17 | Yes | Skin flap | Trans phalangeal | No | Hand |
| 18 | Yes | Skin flap | Chopart | No | Foot |
| 19 | Yes | Skin graft | Pirogoff | No | Foot |
| 20 | Yes | No | No | Fasciotomy | Hand |
| 21 | Yes | No | No | Fasciotomy | Foot |
| 22 | Yes | No | No | Fasciotomy | Foot |
| 24 | Yes | No | No | Synovectomy | Hand |
| 24 | Yes | No | No | Suture for ruptured flexor tendon | Hand |

10% [2, 6, 7], the data from our study were five times higher. Single anatomical site involvement was more common in the SSTMSPS and mixed groups, with no significant differences between the two. Not surprisingly, the lower extremities were the most frequently affected anatomical region, as the legs are either submerged in water contaminated with urine, feces and fuel in the cayuco or else dangling overboard during the crossing [4, 25]. However, up to a quarter of patients with SSTMSPS had upper extremity involvement, which was probably due to the arms being tethered with ropes as the vessels increase in size (from small boats to cayucos), causing injuries like those mentioned above [5]. The most frequent lesions were ulcers, followed by cellulitis. The microbiological studies that were performed showed a discrete increase in gram-negative microorganisms compared to gram-positive cocci, which differs from the general data obtained in all types of skin and soft tissue infections (SSTI) [26]. Among the gram-positive cocci, *S. aureus* was the most frequently isolated, which is consistent with other SSTI series, while the gram-negatives included unusual microorganisms such as *Shewanella algae*, *Morganella morganii*, *Vibrio alginolyticus* or *Aeromonas hydrophila*. This information is crucial for the choice of initial empiric treatment and the prevention of further complications.

Regarding the analytical data, it should be noted that the values considered normal for the native population are not normal for migrants, especially those of sub-Saharan origin [24]. Of note was the presence of anemia in approximately half the patients. Most was characterized as normocytic, although there was a non-negligible percentage of microcytic anemia (16.7%), related to the common hemoglobinopathies in this population [27]. Leukopenia was also very

frequent, found in practically two thirds of the patients. This ethnic leukopenia is due to genetic factors, specifically to alterations in the ACKR1 gene (previously called DARC) [28], which is associated with a decrease in *Plasmodium vivax* infection but does not lead to a higher frequency of bacterial infections [24]. Eosinophilia was observed in only 11.6% of cases, in contrast to another series of immigrants from the same geographical region studied after their arrival in our country, which was related to the presence of helminthiasis [29]. This discrepancy has been attributed to the effect of proinflammatory cytokines on hemopoiesis [30]. Finally, it should be noted that there was no statistical association between the analytical variables and the different forms of patera foot considered.

In this group of patients, it is always important to consider other diseases present before the trip that are more common in their countries of origin or were acquired during travel. With respect to pre-travel infections, we emphasize that our data represent the expected values, based on their countries of origin, for HIV infection, previous syphilis and hepatitis B virus infection, and lower values for tuberculosis and malaria. Although it is not possible to determine the exact time of infection due to the incubation period, we found a high percentage of giardiasis, *Salmonella enterica* infections, but only 5 cases of scabies, which differs from another series in which scabies accounted for 58% of cases [7].

Newly arrived migrants were quarantined until the absence of SARS-CoV-2 infection could be demonstrated. SARS-CoV-2 infection was observed in one fifth of the patients. This is slightly higher than the figure published by Sisti et al [31], possibly because our study period was longer, the patients were younger and the countries of origin were different. On the other hand, the severity of SARS-CoV-2 infection was minimal, which has been explained by the so-called African paradox [32] involving genetic factors (i.e. ACE-2 gene polymorphisms), environmental factors (climate), co-infections (i.e. malaria) and average age.

A finding not reported in other patients with this syndrome but documented in 10% of patients in our series was the presence of extrapulmonary air, specifically spontaneous pneumomediastinum (20 patients) associated with subcutaneous emphysema in 8 cases. This complication usually occurs in young people [33] and males [34]. One of the main causes is frequent vomiting due to increased intrathoracic pressure (such as the Valsalva maneuver). It is well known that patients frequently ingest seawater during the crossing, which causes frequent vomiting and abdominal pain, and could explain our findings. In patients presenting with this complication, we did not observe an increase in mortality, as is usual in spontaneous pneumomediastinum. On the other hand, although the presence of extrapulmonary air has been associated with COVID-19 [35], we did not find a significant association between the presence of pneumomediastinum and concomitant SARS-CoV-2 infection in our series.

The natural evolution of skin, soft tissue and musculoskeletal lesions is poor, which is attributable to the thicker, less extensible skin and the presence of rhabdomyolysis. This causes muscle edema and decreased perfusion leading to ischemia [9, 13] and creates a compartment syndrome-like situation. Indeed, in the first description of 'patera foot' by our group, there were 14% of major amputations [4]. Due to better clinical management (medical and surgical), all of these patients had a good evolution and all amputations were minor [19]. Most patients were treated with amoxicillin/clavulanate, alone or in combination with other antimicrobials (mainly doxycycline). Patients with SSTMSPS were significantly more likely to be admitted to the ICU. The median length of stay was 9 days, as previously described [6], and was, as expected, significantly longer when both syndromes coexisted. In contrast to other series [6], no patient died while admitted.

This study has some limitations, the main ones being the challenges of studying this population. These include the language barrier, which prevents taking a correct anamnesis for knowledge of the underlying pathology (many of them speak only dialects of their own country, such

as Wolof or Bambara), a reluctance to reveal their true origin due to the possibility of deportation because of international agreements, plus cultural or religious factors that affect the collection of samples, and transfers to other centers or places that prevent adequate follow-up [2–8, 20].

## Conclusion

In summary, the data obtained indicate the specific characteristics of *patera syndrome* in newly arrived migrants so that the most effective treatment for optimal outcome can be initiated as early as possible

## Acknowledgments

We thank Janet Dawson for her help in revising the English version of the manuscript. Silvia Rivero-Martel for the design of **Fig 1** and Alejandro Martín-Sánchez MD, Francisco Romero-Santana MD and Raquel Martínez-Goñi MD for their contribution to the management of some patients.

## Author Contributions

**Conceptualization:** Elena Pisos-Álamo, José-Luis Pérez-Arellano.

**Data curation:** Elena Pisos-Álamo, Michele Hernández-Cabrera, Laura López-Delgado, Nieves Jaén-Sánchez, Christian Betancort-Plata, Carmen Lavilla Salgado, Laura Suárez-Hormiga, Marta Briega-Molina, Cristina Carranza-Rodríguez.

**Formal analysis:** Elena Pisos-Álamo, Michele Hernández-Cabrera, Laura López-Delgado, Nieves Jaén-Sánchez, Christian Betancort-Plata, Carmen Lavilla Salgado, Laura Suárez-Hormiga, Marta Briega-Molina, Cristina Carranza-Rodríguez.

**Investigation:** Elena Pisos-Álamo, Cristina Carranza-Rodríguez, Margarita Bolaños-Rivero, Araceli Hernández-Betancor, José-Luis Pérez-Arellano.

**Methodology:** Elena Pisos-Álamo, Michele Hernández-Cabrera, Laura López-Delgado, Nieves Jaén-Sánchez, Christian Betancort-Plata, Carmen Lavilla Salgado, Laura Suárez-Hormiga, Marta Briega-Molina, Cristina Carranza-Rodríguez, Margarita Bolaños-Rivero, Araceli Hernández-Betancor.

**Project administration:** José-Luis Pérez-Arellano.

**Supervision:** José-Luis Pérez-Arellano.

**Validation:** Cristina Carranza-Rodríguez.

**Visualization:** José-Luis Pérez-Arellano.

**Writing – original draft:** Elena Pisos-Álamo, José-Luis Pérez-Arellano.

**Writing – review & editing:** Cristina Carranza-Rodríguez, José-Luis Pérez-Arellano.

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
