## [Decision Letter · Decision Letter 0]

9 Sep 2024

PONE-D-24-30053‘Patera syndrome’ during the COVID-19 pandemic in the Canary Islands (Spain)PLOS ONE

Dear Dr. Carranza-Rodriguez,

Thank you for submitting your manuscript to PLOS ONE. After careful consideration, we feel that it has merit but does not fully meet PLOS ONE’s publication criteria as it currently stands. Therefore, we invite you to submit a revised version of the manuscript that addresses the points raised during the review process.

We look forward to receiving your revised manuscript.

Kind regards,

Felix Bongomin, MB ChB, MSc, MMed, FECMM

Academic Editor

PLOS ONE

Journal Requirements:

1. When submitting your revision, we need you to address these additional requirements. Please ensure that your manuscript meets PLOS ONE's style requirements, including those for file naming. The PLOS ONE style templates can be found at https://journals.plos.org/plosone/s/file?id=wjVg/PLOSOne_formatting_sample_main_body.pdf and https://journals.plos.org/plosone/s/file?id=ba62/PLOSOne_formatting_sample_title_authors_affiliations.pdf 2. Thank you for uploading your study's underlying data set. Unfortunately, the repository you have noted in your Data Availability statement does not qualify as an acceptable data repository according to PLOS's standards. At this time, please upload the minimal data set necessary to replicate your study's findings to a stable, public repository (such as figshare or Dryad) and provide us with the relevant URLs, DOIs, or accession numbers that may be used to access these data.  For a list of recommended repositories and additional information on PLOS standards for data deposition, please see https://journals.plos.org/plosone/s/recommended-repositories. 3. In the online submission form, you indicated that your data will be submitted to a repository upon acceptance.  We strongly recommend all authors deposit their data before acceptance, as the process can be lengthy and hold up publication timelines. Please note that, though access restrictions are acceptable now, your entire minimal  dataset will need to be made freely accessible if your manuscript is accepted for publication. This policy applies to all data except where public deposition would breach compliance with the protocol approved by your research ethics board. If you are unable to adhere to our open data policy, please kindly revise your statement to explain your reasoning and we will seek the editor's input on an exemption. 4. Your ethics statement should only appear in the Methods section of your manuscript. If your ethics statement is written in any section besides the Methods, please move it to the Methods section and delete it from any other section. Please ensure that your ethics statement is included in your manuscript, as the ethics statement entered into the online submission form will not be published alongside your manuscript. 5. We note that Figure 1 in your submission contain [map/satellite] images which may be copyrighted. All PLOS content is published under the Creative Commons Attribution License (CC BY 4.0), which means that the manuscript, images, and Supporting Information files will be freely available online, and any third party is permitted to access, download, copy, distribute, and use these materials in any way, even commercially, with proper attribution. For these reasons, we cannot publish previously copyrighted maps or satellite images created using proprietary data, such as Google software (Google Maps, Street View, and Earth). For more information, see our copyright guidelines: http://journals.plos.org/plosone/s/licenses-and-copyright. We require you to either (1) present written permission from the copyright holder to publish these figures specifically under the CC BY 4.0 license, or (2) remove the figures from your submission: 1. You may seek permission from the original copyright holder of Figure 1 to publish the content specifically under the CC BY 4.0 license.   We recommend that you contact the original copyright holder with the Content Permission Form (http://journals.plos.org/plosone/s/file?id=7c09/content-permission-form.pdf) and the following text:“I request permission for the open-access journal PLOS ONE to publish XXX under the Creative Commons Attribution License (CCAL) CC BY 4.0 (http://creativecommons.org/licenses/by/4.0/). Please be aware that this license allows unrestricted use and distribution, even commercially, by third parties. Please reply and provide explicit written permission to publish XXX under a CC BY license and complete the attached form.” Please upload the completed Content Permission Form or other proof of granted permissions as an ""Other"" file with your submission. In the figure caption of the copyrighted figure, please include the following text: “Reprinted from [ref] under a CC BY license, with permission from [name of publisher], original copyright [original copyright year].” 2. If you are unable to obtain permission from the original copyright holder to publish these figures under the CC BY 4.0 license or if the copyright holder’s requirements are incompatible with the CC BY 4.0 license, please either i) remove the figure or ii) supply a replacement figure that complies with the CC BY 4.0 license. Please check copyright information on all replacement figures and update the figure caption with source information. If applicable, please specify in the figure caption text when a figure is similar but not identical to the original image and is therefore for illustrative purposes only.The following resources for replacing copyrighted map figures may be helpful: USGS National Map Viewer (public domain): http://viewer.nationalmap.gov/viewer/The Gateway to Astronaut Photography of Earth (public domain): http://eol.jsc.nasa.gov/sseop/clickmap/Maps at the CIA (public domain): https://www.cia.gov/library/publications/the-world-factbook/index.html and https://www.cia.gov/library/publications/cia-maps-publications/index.htmlNASA Earth Observatory (public domain): http://earthobservatory.nasa.gov/Landsat: http://landsat.visibleearth.nasa.gov/USGS EROS (Earth Resources Observatory and Science (EROS) Center) (public domain): http://eros.usgs.gov/#Natural Earth (public domain): http://www.naturalearthdata.com/

Additional Editor Comments (if provided):

Reviewers' comments:

Reviewer's Responses to Questions

**Comments to the Author**

1. Is the manuscript technically sound, and do the data support the conclusions?

Reviewer #1: Yes

Reviewer #2: Yes

Reviewer #3: Yes

2. Has the statistical analysis been performed appropriately and rigorously? 

Reviewer #1: No

Reviewer #2: N/A

Reviewer #3: Yes

3. Have the authors made all data underlying the findings in their manuscript fully available?

Reviewer #1: No

Reviewer #2: Yes

Reviewer #3: Yes

4. Is the manuscript presented in an intelligible fashion and written in standard English?

Reviewer #1: Yes

Reviewer #2: Yes

Reviewer #3: Yes

5. Review Comments to the Author

Reviewer #1: The paper describes the called “Patera syndrome” during the COVID-19 pandemic in the Canary Islands (Spain). First, I am agreeing with the title, especially with the “patera patients and syndrome” terms. It is very illustrative and true. The bibliography is actual and appropriate.

It is generally well written and not too long (However, the data should be provided as supplementary material, especially those used for the data analysis). The purpose of the study is well explained in the introduction and appears to be adequately referenced. The conclusions are brief but summarize the obtained results well.

However, the statistical analysis is poor or incomplete. The authors provided a section called “statistical analysis” (see line 198), where the different statistic tests are described and its application. However, the obtained results from the different contrast or situations were not provided or simply commented and discussed in the paper. For instance, in the Table 2 and Table 5, the authors used different test but any information about the results was provided. In the Table 2, only the criteria for APS was significative (p<0.05), whereas it was not significance for the rest of contrast. A similar situation was observed for the Table 5 results. This needs an explanation.

In the Discussion section (see lines 434-448) the authors speak on the analytical data but any information was provided. In this context, the authors comments data in terms of percentages on different clinical situations but no statistical analysis data.

In addition, there is a section called “Analytical data” (see lines 280-285). The authors says “However, there was no significant correlation between determinations obtained at the two time points”. What does this sentence mean? It is very confused. Does this apply only to laboratory data or to all variables used to define PSA? This needs some explanation.

In opinion of this reviewer, the manuscript is suitable for publication in the PLOS ONE Journal after minor revision.

Reviewer #2: This work is a retrospective study of patients who are admitted to a University Hospital after a crossing in patera, arriving to the island of Gran Canaria. The work is well developed and presented, and provides relevant information on this type of patients. We set out below some issues that in our opinion should be clarified.

Major issues:

1- The title should be improved, as it does not only relate to Patera Syndrome, but also to Skin and soft tissue or musculoskeletal patera syndrome (SSTMSPS). Perhaps including “COVID-19” in the title is misleading in terms of the objective of the article, since the data on this disease are very few and of a marginal nature with respect to other pathologies detected.

2- The bibliographic reference of the definition of APS and SSTMSPS is not clear. Is it being defined by the authors for the first time? If so, other conditions should in our opinion be included in APS, such as cardiorespiratory arrest (even if they had no cases in their series). The division of SSTMSPS, from the APS, of which we believe it is a part, should also be justified.

3- Given that probably not all patients who come to the hospital emergency department are admitted, the admission criteria should be included. Is it understood that all patients with PHC criteria were admitted to the ward or did some remain in the Emergency Department and were later discharged?

4- As these were severe cases, was there any out-of-hospital treatment on arrival and prior to admission to the Hospital (serum therapy, etc)?

Minor questions:

1- Line 90. Indicate how many of the migrants from the province of Las Palmas correspond to the island of Gran Canaria, where the study was carried out (therefore not including the islands of Lanzarote and Fuerteventura).

2- Line 134. Is this hospital the reference center for all the migrants on patera or is there another university hospital on the island that could attend them?

3- M&M: did all the patients come to the Hospital directly from the port or did some of them come later, from the interbording center?

4- Discussion:

- Line 344. I think there is an error, as the series only includes data from one of the hospitals in the Canary Islands. If the 56,000 are only from the island of Gran Canaria, this should be made clear in the text. This is important when calculating the % requiring hospitalization, which in any case would be very low.

-Should pneumomedicine be included as a criterion for PHC?

- Did they receive admissions from other islands (Lanzarote and Fuerteventura)?

- It should be noted that, after the study, the pressure of arrivals has moved to the island of El Hierro, where there have been some deaths in hospital care and transfers in critical condition to the referral hospital. (Mora Peces I, Galvez Rodríguez M. Seven nights, Sept nuits. Emergencias. 2024;36:148-148. Mora Peces I, Gálvez Rodríguez M. Health response of El Hierro to maritime migration in 2023. Rev Esp Urg Emerg. 2024;3:1-2)

5- Limitations:

-Possible attendances in other centers on the island of Gran Canaria?

-Missed cases of final diagnosis (line 245), should be included in the limitations and indicate possible reasons for these losses.

6- Tables and figures:

- Figure 3: Distribution of arrivals by month. Does this refer to arrivals on the island of Gran Canaria, the province or the entire Canary archipelago?

- Figure 5. The pneumomediastinum images do not seem to provide new information. We suggest deleting it.

Reviewer #3: The ‘patera syndrome’ during the COVID-19 pandemic in the Canary Islands (Spain)’ is a well-written review of patients treated for “patera syndrome” in a specific significant setting.

Concept, data and presentation of the work will be interesting for the future management of this type of patients.

Minor comments

- If possible, it would be very interesting to complete the text with data on antibiotic treatment, especially initial empirical treatment. A new table in “RESULTS” and a clear conclusion in the last part of the DISCUSSION are recommended.

- The reviewer assumes that there are few specimens for pathological review. But still, if possible, consider adding histological data from the amputation specimens: inflammation only? vasculitic findings?

6. PLOS authors have the option to publish the peer review history of their article (what does this mean?). If published, this will include your full peer review and any attached files.

Reviewer #1: No

Reviewer #2: No

Reviewer #3: No

---

## [Author Response · Author response to Decision Letter 0]

27 Sep 2024

Dear Mr. Bongomin:

 Thank you very much for your comments, as well as those of the three reviewers. A detailed response is attached. Please let us know if we can be of any further assistance.

 Academic editor:

1. With respect to the style requirements, we have adapted the previous manuscript in accordance with the PLOS ONE style templates , both in the “Revised Manuscript with Track Changes” file and in the “Manuscript” file.

2. The 5 figures have been converted to tif format using the Preflight Analysis and Conversion Engine (PACE) digital diagnostic tool.

3. Figure 1 was created by Silvia Rivero-Martel (see Acknowledgements), by adding the migration routes to the map accessible through the web (https://d-maps.com/index.php?lang=es), more specifically https://d-maps.com/carte.php?num_car=3138&lang=es. 

This application is freely available and the conditions of use are attached: 

However, both figure and the references in the text may be deleted if it is deemed necessary.

4. We have reviewed the reference list and confirm that it is complete and correct. 

5. The Ethics statement has been unified and can be found in the Patients and Methods section. 

6. The Repository: this has been uploaded to the stable, open repository Zenodo (DOI: 10.5281/zenodo.13807911).

 Reviewers’ Comments to the Author:

Reviewer #1: 

The paper describes the called “Patera syndrome” during the COVID-19 pandemic in the Canary Islands (Spain). First, I am agreeing with the title, especially with the “patera patients and syndrome” terms. It is very illustrative and true. The bibliography is actual and appropriate. It is generally well written and not too long (However, the data should be provided as supplementary material, especially those used for the data analysis). The purpose of the study is well explained in the introduction and appears to be adequately referenced. The conclusions are brief but summarize the obtained results well. Thank you for your comments.

However, the statistical analysis is poor or incomplete. The authors provided a section called “statistical analysis” (see line 198), where the different statistic tests are described and its application. However, the obtained results from the different contrast or situations were not provided or simply commented and discussed in the paper. For instance, in the Table 2 and Table 5, the authors used different test but any information about the results was provided. In the Table 2, only the criteria for APS was significative (p<0.05), whereas it was not significance for the rest of contrast. A similar situation was observed for the Table 5 results. This needs an explanation. In the Discussion section (see lines 434-448) the authors speak on the analytical data but any information was provided. In this context, the authors comments data in terms of percentages on different clinical situations but no statistical analysis data.

With all due respect, we do not really understand these observations. Both Table 2 and Table 5 give the descriptive results, the statistical tests used and statistical significance. In the Discussion section we have added brief information about the lack of differences between APS and SSTMPS. At the same time, much of the analytical data (i.e. serum ions, osmolality, CK activity, plasma creatinine) are included in the previous section on APS criteria. 

In addition, there is a section called “Analytical data” (see lines 280-285). The authors says “However, there was no significant correlation between determinations obtained at the two time points”. What does this sentence mean? It is very confused. Does this apply only to laboratory data or to all variables used to define PSA? This needs some explanation.

We fully agree with the reviewer that the text is confusing and has been modified to make it clearer. What we wanted to say was that the hematimetric and biochemical data in particular show notable differences between the initial samples obtained and those taken at 72h (see Repository). Therefore, we used the initial analytical data to define APS, but then compared the data later when the situation was more stable (72h).

In opinion of this reviewer, the manuscript is suitable for publication in the PLOS ONE Journal after minor revision. Thank you once again for your comments .

Reviewer #2: 

This work is a retrospective study of patients who are admitted to a University Hospital after a crossing in patera, arriving to the island of Gran Canaria. The work is well developed and presented, and provides relevant information on this type of patients. Thank you for your comments.

We set out below some issues that in our opinion should be clarified.

Major issues:

 1- The title should be improved, as it does not only relate to Patera Syndrome, but also to Skin and soft tissue or musculoskeletal patera syndrome (SSTMSPS). 

We agree with reviewers 1 and 3 that “Patera syndrome” is appropriate since it serves as an umbrella term for both Acute Patera Syndrome and SSTMSPS

Perhaps including “COVID-19” in the title is misleading in terms of the objective of the article, since the data on this disease are very few and of a marginal nature with respect to other pathologies detected. 

We think that the reference to the COVID-19 pandemic is relevant for at least two reasons: a) it was one of the factors that triggered the sharp increase in irregular migration from Africa to the Canary Islands, and b) hospitals in the Canary Islands, and specifically in Gran Canaria, were overwhelmed by the number of patients. 

2- The bibliographic reference of the definition of APS and SSTMSPS is not clear. Is it being defined by the authors for the first time?

Yes, the definitions of Patera syndrome, APS and SSTMPS are the authors’ own. 

 If so, other conditions should in our opinion be included in APS, such as cardiorespiratory arrest (even if they had no cases in their series). 

Previous cardiorespiratory arrest would already be included in the defined criteria . 

The division of SSTMSPS, from the APS, of which we believe it is a part, should also be justified.

The two syndromes are not mutually exclusive; in fact, this is specifically noted in the text. 

3- Given that probably not all patients who come to the hospital emergency department are admitted, the admission criteria should be included. 

As this was a retrospective study, the criteria for admission and referral were the usual emergency department criteria..

4- As these were severe cases, was there any out-of-hospital treatment on arrival and prior to admission to the Hospital (serum therapy, etc)?

Obviously, on arrival, the patients were treated with the conventional measures for control of dehydration, hypothermia, shock and wound care, among other things, and then referred to hospital if more serious data were observed. 

Minor questions:

1- Line 90. Indicate how many of the migrants from the province of Las Palmas correspond to the island of Gran Canaria, where the study was carried out (therefore not including the islands of Lanzarote and Fuerteventura).

It is difficult to determine exactly how irregular migrants arriving in the Canary Islands by sea were distributed by island /province, On the one hand, there are no accessible official data (such as those from the Ministry of the Interior) and on the other, those that are available vary according to source and time period. Thus, according to the Red Cross, in the year 2020 (January 1 to December 31) (https://www2.cruzroja.es/web/ahora/inmigracion-canarias#), the places with the highest number of migrants were Gran Canaria, Tenerife and Fuerteventura, receiving 16,463, 3,669, and 1,394 persons respectively. However newspaper data from 2023 indicate that 7,800 migrants arrived on the island of Gran Canaria and 3,250 on Lanzarote 

 (https://www.elindependiente.com/espana/2024/01/03/ and https://www.lancelotdigital.com/lanzarote/). Finally, as you point out in the reference (Mora Peces I, Gálvez Rodríguez M. Health response of El Hierro to maritime migration in 2023. Rev Esp Urg Emerg. 2024;3:1-2), the pressure of arrivals has shifted to the island of El Hierro, where there have been some deaths in hospital care and transfers in critical condition to the referral hospital (in this case on the island of Tenerife). 

2- Line 134. Is this hospital the reference center for all the migrants on patera or is there another university hospital on the island that could attend them.

El Complejo Hospitalarario Universitario Insular Materno Infantil is the reference center on the island for obvious reasons, including the fact that it covers the south of Gran Canaria. However, during the study period, when the hospital was saturated with patients with Covid-related needs, some patients were transferred to the other University Hospital on the island (Dr. Negrín).

3- M&M: did all the patients come to the Hospital directly from the port or did some of them come later, from the interbording center?

Already answered previously.

4- Discussion:

-Line 344. I think there is an error, as the series only includes data from one of the hospitals in the Canary Islands. If the 56,000 are only from the island of Gran Canaria, this should be made clear in the text. This is important when calculating the % requiring hospitalization, which in any case would be very low.

Indeed, this was due to the difficulty in obtaining data, and we have now clarified this in the new version. 

-Should pneumomedicine be included as a criterion for PHC?

We think not, because although they are relatively common, they are neither sensitive nor specific. 

- Did they receive admissions from other islands (Lanzarote and Fuerte-ventura)?

 None of the admissions included patients from the other islands (Lanzarote and Fuerteventura) in the province of Las Palmas.

5- Limitations:

-Possible attendances in other centers on the island of Gran Canaria

Already answered.

-Missed cases of final diagnosis (line 245), should be included in the limitations and indicate possible reasons for these losses.

In 9 patients, a definitive diagnosis could not be made despite the tests carried out. This information is included in the limitations of the study. Any interpretation of this data is speculative. 

.

6- Tables and figures: 

- Figure 3: Distribution of arrivals by month. Does this refer to arrivals on the island of Gran Canaria, the province or the entire Canary archipelago?

The results are for the Canary Islands archipelago; specific data were not available, as previously stated. 

- Figure 5. The pneumomediastinum images do not seem to provide new information. We suggest deleting it.

None of the images of extrapulmonary air actually provide new radiological information; we thought it was interesting to attach them in the context of this syndrome. Nevertheless, if you consider it appropriate, this figure can be deleted.

Reviewer #3: 

The ‘patera syndrome’ during the COVID-19 pandemic in the Canary Islands (Spain)’ is a well-written review of patients treated for “patera syndrome” in a specific significant setting. Concept, data and presentation of the work will be interesting for the future management of this type of patients.

Minor comments

- If possible, it would be very interesting to complete the text with data on antibiotic treatment, especially initial empirical treatment. A new table in “RESULTS” and a clear conclusion in the last part of the DISCUSSION are recommended.

Empirical antimicrobial treatment varied a good deal depending, among other things, on the extent and depth of the lesions and the prescribing physician. We have added a new table (6) and a mention of this in the discussion. 

- The reviewer assumes that there are few specimens for pathological review. But still, if possible, consider adding histological data from the amputation specimens: inflammation only? vasculitic findings?

Unfortunately, we do not have histopathologic data on the amputation specimens.

---

## [Editor Report · Decision Letter 1]

7 Oct 2024

‘Patera syndrome’ during the COVID-19 pandemic in the Canary Islands (Spain)

PONE-D-24-30053R1

Dear Dr. Carranza-Rodriguez,

We’re pleased to inform you that your manuscript has been judged scientifically suitable for publication and will be formally accepted for publication once it meets all outstanding technical requirements.

Kind regards,

Felix Bongomin, MB ChB, MSc, MMed, FECMM

Academic Editor

PLOS ONE
---

## [Editor Report · Acceptance letter]

13 Oct 2024

PONE-D-24-30053R1 

PLOS ONE

Dear Dr. Carranza-Rodriguez, 

I'm pleased to inform you that your manuscript has been deemed suitable for publication in PLOS ONE. Congratulations! Your manuscript is now being handed over to our production team.

Kind regards, 

on behalf of

Dr. Felix Bongomin 

Academic Editor

PLOS ONE